# Behavioral and Psychological Symptoms of Dementia: Prevalence, Symptom Severity, and Caregiver Distress in South-Western Uganda—A Quantitative Cross-Sectional Study

**DOI:** 10.3390/ijerph20032336

**Published:** 2023-01-28

**Authors:** Ronald Kamoga, Vincent Mubangizi, Judith Owokuhaisa, Moses Muwanguzi, Sylivia Natakunda, Godfrey Zari Rukundo

**Affiliations:** 1Department of Anatomy, Mbarara University of Science and Technology, Mbarara P.O. Box 1410, Uganda; 2Department of Community Practice and Family Medicine, Mbarara University of Science and Technology, Mbarara P.O. Box 1410, Uganda; 3Department of Physiotherapy, Mbarara University of Science and Technology, Mbarara P.O. Box 1410, Uganda; 4Faculty of Medicine, Mbarara University of Science and Technology, Mbarara P.O. Box 1410, Uganda; 5Department of Psychiatry, Mbarara University of Science and Technology, Mbarara P.O. Box 1410, Uganda

**Keywords:** behavioral and psychological symptoms of dementia, dementia, distress, neuropsychiatric symptoms, caregiver, Uganda

## Abstract

The purpose of the study was to investigate behavioral and psychological symptoms (BPSD) prevalence, severity, and distress experienced by caregivers of people living with dementia (PLWD). A cross-sectional, population-based study was conducted in a rural area in southwestern Uganda. A Neuropsychiatric Inventory Questionnaire (NPI-Q) was used to determine the presence of BPSD as perceived by caregivers of PLWD. We carried out both descriptive and inferential data analysis. A total of 175 caregivers of PLWD were enrolled in this study. Among PLWD, 99% had presented BPSD in the past month. Hallucinations (75%) and dysphoria/depression (81%) were the two BPSD that occurred most frequently. Most participants (70%) stated that PLWD experienced hallucinations of significant severity. Aberrant motor activity was reported by 60% of the participants as the type of BPSD that caused severe distress. There was a high positive correlation (0.82) between the total severity score and total distress scores. Interventions aimed at addressing dysphoria and hallucinations may be essential for the reduction of caregiver distress. These findings point to the need for promoting early screening for BPSDs and the provision of support to caregivers.

## 1. Introduction

Dementia is a clinical syndrome characterized by an irreversible decline in cognitive functioning. It is a major public health concern that affects mankind globally, and its prevalence has increased partly because the world’s population has gotten older in general [1,2,3,4]. Alzheimer’s disease is the most common cause of dementia [5]. It is estimated that 55 million people have dementia globally, and the prevalence is anticipated to reach 152 million by 2050 [6]. About 2.13 million people suffer from Alzheimer’s disease in sub-Saharan Africa [3]. A study carried out among persons aged 60 years and above in southwestern Uganda found the prevalence of dementia and related diseases at 20% [7], which was about three times higher than the 7.2% prevalence estimate in sub-Saharan Africa [3].

About 90% of all people living with dementia (PLWD) will develop one or more symptoms of BPSD over the course of their illness [8]. Even in the early stages of cognitive impairment, neuropsychiatric symptoms are frequent, with estimated rates of 35–85% in subjects with mild cognitive impairment [8,9]. The International Psychogeriatric Association regards behavioral and psychological symptoms of dementia (BPSD) as signs and symptoms of disturbed thought content, perception, behavior, or mood [10]. Behavioral and psychological symptoms of dementia (BPSD) refer to the spectrum of non-cognitive and non-neurological symptoms of dementia [8]. Agitation, aberrant motor behavior, anxiety, elation, irritability, depression, apathy, disinhibition, delusions, hallucinations, and changes in sleep or appetite are all symptoms of BPSD [8,11]. BPSDs tend to occur in clusters that can vary by time, severity, size, and diagnosis. There is a certain degree of concordance in the groups of BPSD, resulting in four syndromes: (1) hyperactivity cluster (agitation, aggression, euphoria, disinhibition, irritability, and aberrant motor activity); (2) psychosis cluster (hallucinations and delusions); (3) *mood liability cluster* (depression and anxiety); and (4) instinctual cluster (appetite disturbance, sleep disturbance, and apathy) [12]. 

In the older Nigerian population, the most commonest reported BPSDs include apathy, nighttime behavior, aberrant motor behavior, agitation, and irritability [13], whereas depression, anxiety, and irritability were the commonest among the Central African Republic and the Democratic Republic of Congo [14]. Evidence shows that BPSD causes significant distress to caregivers, which compromises the quality of life (QoL) of people living with dementia (PLWD) [15,16]. In patients with dementia, depressive symptoms are associated with worse self-reported QoL scores [17], whereas mood and psychotic symptoms predict changes in the QoL two years later [18]. Additionally, an increased number of BPSD correlates negatively with survival rates over a 3-year period [19,20]. The presence of psychosis in dementia is associated with increased mortality and hastening cognitive decline [21,22]. BPSD contributes to caregiver burnout and distress as well as increasing costs of care for PLWD [8,23,24,25].

The cure for dementia has remained elusive; only supportive treatment is offered [26]. The support care given to PLWD in Africa is mainly informally provided by family members or the significant other of the person incapacitated by an illness, with or without support from the government [27,28]. The care provided includes household chores, meal preparation and feeding, bathing, dressing, transportation, and medications depending on the severity of the illness. Thus, caregivers of people with dementia, often called the “invisible second patients”, are critical to the care recipients’ quality of life (QoL). Although often beneficial, being a caretaker generally has negative repercussion, high burden and psychological morbidity rates as well as social isolation, physical illness, and financial hardships [29].

While some neuropsychiatric symptoms can be more often recognized in a specific pathological subtype of dementia, the clinical presentation has a wide variation within each subtype and even within each individual suffering from dementia [20,30]. Management of BPSD is a key component of a comprehensive approach to the treatment of dementia, requiring a careful combination of pharmacological and non-pharmacological interventions. Treatment of these symptoms remains problematical, with an increased risk of psychotropic medication misuse, and thus, this represents an important challenge for clinicians. A systematic review of 15 guidelines for managing BPSD laid out in 2012 showed that non-pharmacological interventions were recommended as a first-line treatment, followed by the least harmful medication for the shortest time possible [31].

BPSD in Uganda has not been well studied. Information about the prevalence and major types of BPSD as well as their impact on caregivers is scanty. Apart from expanding the body of knowledge related to dementia in Uganda, this study generates novel information about BPSD symptoms and informs policymaking and clinical practice interventions regarding the prevention and management of BPSD symptoms. Therefore, this study sought to determine the prevalence, types, and impact of BPSD symptoms on caregivers in southwestern Uganda. We hope that this research will contribute helpful information for a better understanding of BPSD, which is useful when caring for PLWD.

## 2. Materials and Methods

### 2.1. Study Design

This was a cross-sectional, population-based, quantitative study.

### 2.2. Study Setting

This study was conducted in June 2022 in Rukiga and Rubanda Districts in the Kigezi Region in rural Southwestern Uganda. The Kigezi Region has been found to have 6% of all older persons (60 years and above) in Uganda, which makes it the sub-region with the highest share of older persons compared to all other sub-regions of Uganda. About 5% of the overall population in the area is 60 years of age or older [32]. With a prevalence rate of 20% among those 60 years of age and above, dementia is highly prevalent in this area [7].

### 2.3. Study Participants

The study participants were caregivers of PLWD who were in care at the two selected health centers, namely, Reach One Touch One Ministries (ROTOM) health center in Rukiga and Heal Medical Centre in Rubanda. ROTOM is a non-governmental organization that reaches out to older people in their care. Heal Medical Centre is a private not-for-profit health facility offering health care services in communities, including care for PLWD. These organizations keep a record of elderly people and the diagnoses of the condition(s) they have. The PLWD are cared for as outpatients, and they are only admitted when there is a condition warranting such action. In this study, a caregiver was defined as a family member or relative who has been helping a PLWD while living with them for at least 6 months. The caregiver had to be 18 years of age or above or an emancipated minor so that s/he could give consent. Caregivers would be excluded if they were under the age of 18 or had mental or cognitive impairments that would make it difficult for them to communicate with the research team. There were no inclusion or exclusion standards for patients with dementia.

### 2.4. Sample Size Determination, Sampling Procedure, and Recruitment

The sample size was determined using the Keish–Leisle formula [33].
(1)$$N=\frac{Z\alpha^{\;2}\;P\left(1-P\right)}{e^{2}}$$
(2)$$N=\frac{{\left(1.96\right)}^{\;2}\;0.884\;\left(1-0.884\right)}{{\left(0.05\right)}^{2}}$$
*N* = 158 + (10% non-response) = 174 participants(3)
where *Zα* is the standard normal distribution of 1.96, which corresponds with a 95% confidence level; *P* is 88.4% the prevalence of at least one BPSD among older persons in a community-based study in rural Tanzania [34]; *e* is the margin of error of 0.05. We added a 10% non-response rate, and the final calculated minimum sample size was 174 participants. It was assumed that the prevalence in the study area would be similar to that in rural Tanzania due to similarities in the study settings.

After being briefed on the goals, objectives, and eligibility requirements of the study, the administrators created lists. Participants were randomly selected from the lists of caregivers of PLWD provided by the administrators of ROTOM in the Rukiga District and Heal Medical Centre in the Rubanda District. Potential participants that met the inclusion criteria were contacted by the administrators, who also requested permission to share their contact information with the research team. Participants were contacted by the research team, and appointments to collect data at home were scheduled. Community entry was led by village health team members who guided the home visit in the different villages served by the two facilities. While in the homes, informed consent was obtained, and interviews were conducted in a private place identified within the homestead.

### 2.5. Data Collection Tools and Data Collection Methods

#### 2.5.1. Neuropsychiatric Inventory–Questionnaire (NPI-Q)

We used the Neuropsychiatric Inventory–Questionnaire (NPI-Q), which was developed and cross-validated with the standard Neuropsychiatric Inventory (NPI), to provide a brief assessment of BPSD [35]. The NPI-Q was adapted from the NPI, a validated informant-based interview that assesses neuropsychiatric symptoms over the previous month [36]. NPI-Q rates symptoms in 12 domains: hallucinations, delusions, agitation/aggression, anxiety, depression, elation/euphoria, apathy, disinhibition, irritability, aberrant motor behavior, sleep/night behavioral disturbance, and eating/appetite changes. The NPI-Q can be a self-administered questionnaire or carried out as an interview completed by informants about PLWD for whom they care. It can be completed in between five to ten minutes.

Each domain begins with a screening question, for example, “Does the patient appear to feel too good or act excessively happy?” The response is either “Yes” (present) or “No” (absent). The informant moves on to the following question if the screening domain question has a negative response. If “Yes”, the informant then rates both the severity of the symptoms present within the last month on a 3-point scale and the ensuing caregiver distress using a 5-point scale. The NPI-Q rates symptom severity and distress of each symptom reported, and total severity and distress scores reflect the sum of individual domain scores. The NPI-Q assigns each symptom a severity score: 1 for mild symptoms (noticeable but not a significant change), 2 for moderate symptoms (significant but not a dramatic shift), and 3 for severe symptoms (very marked or prominent, a dramatic change). This is how distressed people are rated: A score of 0 indicates no distressing experiences, a score of 1 indicates minimal distressing experiences, a score of 2 indicates mild distressing experiences, a score of 3 indicates moderate distressing experiences, a score of 4 indicates severe distressing experiences, and a score of 5 indicates extremely distressing experiences (extremely distressing, unable to cope). We took severity to be significant if it was moderate or severe (scores ≥ 2) since mild severity, though noticeable, was not a significant change. Distress was considered significant if it was moderate, severe, or extreme/very severe (scores ≥ 3) since it was considered to be generally easy to cope with distress below a score of 3.

#### 2.5.2. Demographic Questionnaire

A study questionnaire was administered to elicit variables such as age, gender, level of education attained, marital status, smoking history, and history of alcohol use. Each variable was assessed by either being present (yes) or absent (no). Study participants’ ages were a self-stated number of years completed. The number of years spent in school was used to determine a person’s level of education. Examples include: never attended school, 1–7 years (primary school), 8–11 years (secondary school), and 12 or more years (tertiary education). Marital status was taken as married/cohabiting, Widow/widower, divorced/separated, or never married. Self-reported information was used to elicit a person’s history of cigarette and alcohol usage. 

Additionally, we asked about past histories of chronic conditions such as type I diabetes mellitus and hypertension. We also elicited employment status as previously informally employed and currently not active, previously informally employed and currently still active, previously formally employed, retired but currently still active, and previously formally employed and retired but currently not active.

Survey questions of all tools were written in English, translated from English into Rukiga-Runyankore, and then back-translated to verify fidelity to the original wording. All study tools were interviewer-administered in the local language (Rukiga-Runyankore) to ensure appropriateness and thorough understanding.

### 2.6. Data Management and Data Analysis

Data were entered into an Excel spreadsheet before being exported to STATA V.15 for analysis. Means and standard deviations were used to describe continuous variables with normal distribution, and percentages were used to describe categorical variables. The Pearson correlation coefficient was used to ascertain the relationships between symptom total symptom severity, total caregiver distress, the cumulative number of BPSD, and other variables. The significant level was at less than 5% for a 95% confidence interval.

### 2.7. Quality Assurance

Quality assurance was implemented at several levels. Data were collected by research assistants who had prior experience conducting surveys. The research assistants were trained on study objectives and using the survey tools and how to conduct themselves in the field. Research assistants participated in a pre-testing of the tools in a different community before the start of data collection. Following data collection, extensive data cleaning was performed to identify and correct dating inconsistencies, missing values, and other issues.

## 3. Results

### 3.1. Socio-Demographic Characteristics of the Study Sample

A total of hundred and seventy-five (175) participants were interviewed, yielding a 100% response rate. Table 1 provides a summary of socio-demographic characteristics. The total percentages may not add up to 100 due to rounding off. Most of the respondents were females (75%). Participants’ average age was 47 (SD 16) years. The average age of females was 46 (SD 16) years, while that of males was 50 (SD 17) years. The majority (73%) were married or cohabiting, and 23% had a positive history of ever using alcohol (47% and 15% of males and females, respectively). 

### 3.2. Prevalence of BPSD

Almost all the 175 caregivers (99%) of PLWD reported that their patient/relative had at least one BPSD. About 5% of participants reported three symptoms, as indicated in Figure 1. About seven percent of participants reported that the PLWD had experienced all 12 symptoms of BPSD. The total mean number of BPSD experienced by each participant was 7 ± 3. Table 2 shows the prevalence of each type of BPSD, mean severity and distress, and frequency of significant severity and distress per symptom. Participants most frequently reported BPSD of dysphoria/depression (81%), hallucinations (75%), anxiety (67%), and appetite/eating (67%). Caregivers were least likely to report disinhibition as a type of BPSD (41%).

The average score for severity was highest for aberrant motor symptoms at 2.6 ± 0.6 and least for appetite/eating at 1.8 ± 0.7. On the other hand, most participants reported experiencing significant severity in hallucinations symptoms (70%). Only 35% of caregivers reported significant severity in elation/euphoria symptoms. The symptoms of hallucinations, disinhibition, and irritability/lability had the greatest mean distress score (2.9 ± 1.7), while elation/euphoria had the lowest (1.6 ± 2.0). Sixty percent of the participants identified the form of BPSD that produced the most significant distress as aberrant motor activity. Few participants (16%) reported significant distress in elation/euphoria symptoms. 

### 3.3. Correlations between the Total Numbers of BPSD with Other Studied Variables

There was a high positive correlation between the total severity score and total distress scores of 0.82, as shown in Table 3. The cumulative number of BPSD exhibited high and very high positive significant associations with the total distress score and severity score, respectively. This implied that caregivers of PLWD with many symptoms experienced severe symptoms and worsening distress. The district of residence of participants showed a significant negative correlation with the number of BPSD although negligible. Gender was also a negatively correlated number in BPSD though the correlation was not significant.

## 4. Discussion

Socio-demographics have an impact on the perceived distress and care burden. In our sample study, the majority of the caregivers were females. This result is consistent with that of the earlier study conducted in India, which revealed that females felt more burdened and distressed by caring for PLWD than males did [37]. Compared to males, females are more empathetic and emotional and hence more likely to be stressed, leading to depression [38]. We did not document the relationship of caregivers to PLWD, but in this rural setting, it is expected that informal caregivers were family members. The level of perceived stress experienced may be affected by how the caregiver is expected to care for PLWD and his or her relationship with the patient [28].

In this rural population-based study in southwestern Uganda, we estimated the prevalence of BPSD at 99%, which was in agreement with a study performed by Mukherjee and others in India, which found a prevalence of 99.1% [20]. This was a bit close to the prevalence of 89% obtained in a study on PLWD related to Parkinson’s disease in which the researchers used an NPI survey instrument [39]. Validation of NPI-Q against the NPI instrument showed that the prevalence of symptoms varied by five percent, while ratings for moderate or severe symptoms differed by less than two percent [35]. Studies indicate that nearly all PLWD will develop at least one BPSD in the course of their disease progression [8,40]. Our estimated prevalence rate of BPSD is higher than the estimated prevalence of 35–85% previously reported although the majority of these studies used a different survey tool, were conducted in high-income countries, and included PLWD who had mild cognitive impairment and were institutionalized [8,9,41]. Moreover, our study was conducted shortly after the lifting of total lockdown due to the coronavirus SARS-cov2 pandemic, which has been shown to cause people to be stressed, and this could have influenced our findings [42,43,44,45]. The prevalent neuropsychiatric symptoms make aspects of care costly, complex, and stressful. This will negatively affect the quality of life, reduce the income of caregivers, and contribute to poor health outcomes for PLWD [11].

A study conducted in the United States of America using the NPI tool after PLWD were confirmed as dementia patients by a committee of expert neurologists and psychiatrists showed that about 45% of participants reported two or fewer symptoms, and 56% of participants reported three or fewer symptoms [46]. This contrasts sharply with our study findings, where only six percent reported two or fewer symptoms, and 11% had three or fewer symptoms. An international multicenter clinical trial with Parkinson’s dementia in Europe and Canada found that 60% of participants had three or more symptoms [39]. The average number of symptoms in our study is about twice the number reported previously [39]. Pharmacotherapy likely helped the participants in highly developed countries by reducing the prevalence and frequency of BPSD and alleviating their symptoms. 

Studies are inconsistent on the most frequent types of BPSD. Past studies have demonstrated that BPSD in PLWD is unpredictable and heterogeneous in the presentation of thought content, emotional experience, motor function, and perception, which may explain the various findings of research on the prevalence of BPSD. For instance, a systematic review performed in 2015 including 48 articles showed that the most common type of BPSD was apathy, and the least prevalent was euphoria [47]. However, reviewers noted that the quality of included studies was not always optimal, and there was significant heterogeneity of prevalence estimates across studies. Our current study showed that the most frequent type of BPSD was dysphoria/depression (81%), and the least frequent was disinhibition (41%). A study undertaken by Baharudi and others in 2019 in home settings in Malaysia found that the most frequent types of BPSD were irritability (84%), apathy (81%), agitation (77%), and appetite (59.4%) [48]. The differences in patterns of BPSD may be due to the tool used, the severity of dementia, environmental parameters, and the number of symptoms studied.

The study showed that BPSD was severe and caused significant distress, which is consistent with other studies [49,50,51,52,53,54]. Caregivers were distressed by BPSD, and it had a significant stress effect on the life of both caregivers and PLWD; it affects their personal and social well-being and makes it more demanding to provide care to PLWD. Caregivers experience most stress from symptoms of depression and anxiety [49,50,51,52,53,54]. The perceived severe distress calls for attention to support and manage these symptoms. This is a challenging task in a resource-constrained country. Studies have shown that caregivers may seek solace in substance abuse, such as marijuana usage and increased alcohol usage [49]. Further studies should be performed to understand the associated factors with severe perceived distress.

### Study Limitations

This study had some limitations. Due to limited resources, we were unable to independently confirm the diagnosis and subtypes of dementia and had to rely solely on the information that was supplied. However, different subtypes of dementia may manifest differently in BPSD [30,55]. The responses of study participants were not cross-checked against any record, such as medical records or national identification cards, and could have been influenced by recall bias and social desirability. We were cognizant of the strong social and cultural values held in the study area about elderly people, which could influence participants to provide socially and culturally acceptable responses about how they were managing. We minimized this by explaining and emphasizing to participants that we wanted them to tell us what was happening, we were not judging them, and that information given would be kept confidential. We did not attempt to find out the duration of caregiving and the condition of the PLWD, which could affect the extent of the BPSDs experienced. In the future, researchers should diagnose the type of dementia their participants have and whether or not they are receiving BPSD treatment.

## 5. Conclusions

BPSD is frequent and significant, causing distress among caregivers. Interventions aimed at addressing them may be essential for the reduction of caregiver distress. These findings point to the need to promote early screening for BPSDs and provide support to caregivers. 

## Figures and Tables

**Figure 1 ijerph-20-02336-f001:**
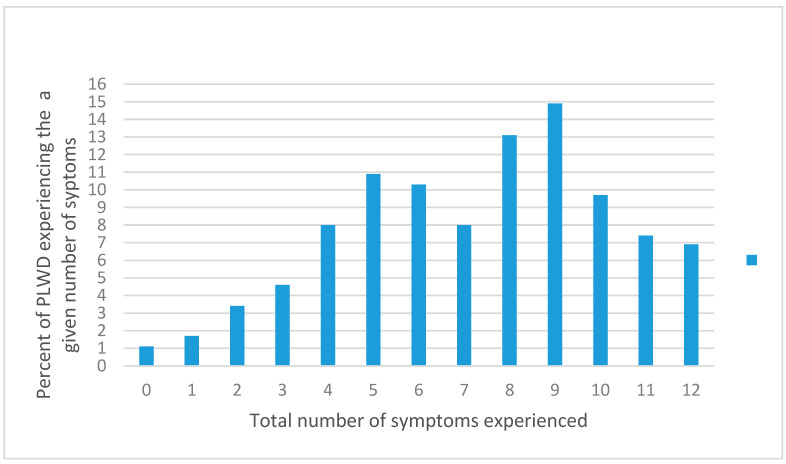
Frequency of behavioral and psychological symptoms of dementia among people living with dementia.

**Table 1 ijerph-20-02336-t001:** Socio-demographic characteristics of participants.

Study Variables	Total Sample; n (%)	Mean ± SD of BPSD
Socio-demographic variables		
District location		
Rubanda District	131 (74.9)	7.7 ± 2.9
Rukiga District	44 (25.1)	5.9 ± 2.5
Age (in completed years)		
60 years and below	148 (84.6)	7.3 ± 3.0
Above 60 years	27 (15.4)	7.3 ± 2.8
Gender		
Female	132 (75.4)	7.6 ± 2.9
Male	43 (24.6)	6.3 ± 2.9
Marital status		
Separated/Divorced	14 (8.0)	7.4 ± 2.4
Married/Cohabiting	128 (73.1)	7.2 ± 3.1
Single	13 (7.4)	6.6 ± 2.7
Widow/Widower	20 (11.4)	7.8 ± 2.6
Level of education		
None	32 (18.3)	7.2 ± 2.9
Primary	114 (65.1)	7.3 ± 2.9
Secondary	19 (10.9)	8.1 ± 3.3
Tertiary	10 (5.7)	6.0 ± 2.2
Employment status		
Previously employed, retired, but currently still active	1 (0.6)	2.0 ± 0
Previously employed, retired, but currently not active	10 (5.7)	6.1 ± 2.0
Previously not employed and currently not active	49 (28.0)	7.4 ± 2.9
Previously not employed and currently still active	115 (65.7)	7.4 ± 3.0
Presence of chronic illness		
Yes	67 (38.3)	7.4 ± 2.7
No	108 (61.7)	7.2 ± 3.1
History of alcohol use		
Yes	40 (22.9)	7.4 ± 2.8
No	135 (77.1)	7.2 ± 3.0
History of cigarette smoking		
Yes	14 (8.0)	7.2 ± 3.0
No	161 (92.0)	7.9 ± 2.3

Abbreviations used: BPSD, behavior and psychological symptoms of dementia; SD, standard deviation.

**Table 2 ijerph-20-02336-t002:** Prevalence of each type of BPSD, mean severity and distress, and frequency of significant severity and distress per symptom as reported by a caregiver.

BPSD Types	Persons Reporting BPSD N = 175 (%)	Mean Severity ± SD	Severity > 2, n (%)	Mean Distress ± SD	Distress > 3, n (%)
Delusions	109 (62.3)	2.4 ± 0.7	95 (54.3)	2.7 ± 1.7	59 (33.7)
Hallucinations	132 (75.4)	2.5 ± 0.6	123 (70.3)	2.9 ± 1.7	76 (43.4)
Agitation/Aggression	90 (51.4)	2.2 ± 0.7	75 (42.9)	2.7 ± 1.6	46 (26.3)
Dysphoria/Depression	142 (81.1)	2.3 ± 0.7	119 (68.0)	2.6 ± 1.7	69 (39.4)
Anxiety	118 (67.4)	2.2 ± 0.6	105 (60.0)	2.7 ± 1.7	61 (34.9)
Elation/Euphoria	94 (53.7)	2.1 ± 0.7	61 (34.9)	1.6 ± 2.0	28 (16.0)
Apathy/Indifference	99 (56.6)	2.3 ± 0.8	82 (46.9)	2.2 ± 1.8	39 (22.3)
Disinhibition	71 (40.6)	2.2 ± 0.8	53 (30.3)	2.9 ± 1.9	39 (22.3)
Irritability/Lability	97 (55.4)	2.4 ± 0.7	87 (49.7)	2.9 ± 1.7	57 (32.6)
Aberrant motor	111 (63.4)	2.6 ± 0.6	105 (60.0)	2.5 ± 1.7	105 (60.0)
Nighttime behavior	90 (51.4)	2.4 ± 0.8	71 (40.6)	2.6 ± 1.9	46 (26.3)
Appetite/Eating	118 (67.4)	1.8 ± 0.7	77 (44.0)	2.1 ± 1.8	49 (28.0)

Abbreviations used: BPSD, behavior and psychological symptoms of dementia; SD, standard deviation; n, number; N, total number.

**Table 3 ijerph-20-02336-t003:** Correlations between the total number of BPSD and other variables studied.

Study Variables	Pearson Correlation Coefficients											
	1	2	3	4	5	6	7	8	9	10	11	12
Age # (1)	1											
District location (2)	0.14	1										
Gender (3)	0.12	0.01	1									
Level of education (4)	−0.33 **	−0.16 *	0.24 **	1								
Marital status (5)	0.12	0.09	−0.1	−0.09	1							
Employment status (6)	−0.05	0.07	−0.20 **	−0.47 **	0.06	1						
Presence of chronic illness (7)	0.33 **	0.22 **	−0.04	−0.17 *	0.16 *	−0	1					
History of cigarette smoking (8)	0.17 *	−0.07	0.08	−0.1	0.08	0.1	−0.06	1				
History of alcohol use (9)	0.07	−0.03	0.32 **	−0.01	−0.1	0	−0.06	0.34 **	1			
Total distress score # (10)	−0.11	−0.22 **	−0.18	0.03	−0	0.1	0.09	0.08	0.03	1		
Total severity score # (11)	−0.11	−0.24 **	−0.23 **	0.01	0.02	0.1	0.08	0.06	0.02	0.82 **	1	
Cumulative number of BPSD # (12)	−0.07	−0.28 **	−0.18	−0.02	0.02	0.1	0.05	0.07	0.02	0.71 **	0.93 **	1

Note: * *p* < 0.05; ** *p* < 0.01; # denotes a continuous variable r2 value represented as: very high correlation positive (negative) = 0.90 to 1.00 (−0.90 to −1.00); high positive (negative) correlation = 0.70 to 0.90 (−0.70 to −0.90); moderate positive (negative) correlation = 0.50 to 0.70 (−0.50 to −0.70); low positive (negative) correlation = 0.30 to 0.50 (−0.30 to −0.50); negligible correlation = 0.00 to 0.30 (.00 to −0.30). Categorical variables were coded as follows: Variable 2 (0 = Rubanda, 1 = Rukiga); 3 (0 = female, 1 = male); 4 (0 = none, 1 = primary, 2 = secondary, 3 = tertiary); 5 (0 = divorced/separated, 1 = married/cohabiting, 2 = single, 3 = widow/widower); 6 (0 = previously employed, retired, but currently still active; 1 = previously employed, retired, but currently not active; 2 = previously not employed and currently inactive; 3 = previously employed and still working); 7–9 (0 = no, 1 = yes).

## Data Availability

The data for the current study are available from the corresponding author upon reasonable request.

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
