# Peer review of "Behavioral and Psychological Symptoms of Dementia: Prevalence, Symptom Severity, and Caregiver Distress in South-Western Uganda—A Quantitative Cross-Sectional Study"

_ijerph, 2023, doi:10.3390/ijerph20032336_

Round 1
Reviewer 1 Report
The topic is interesting and the results could be useful as starting point for further research. However, there are some aspects in which it would be convenient to deepen, since the detail and precision are lacking in the explanation (highlighted in the paper). On the other hand, references to studies from other countries should be more supported in the explanation, to be sure they are fully applicable.

Reviewer 2 Report
The manuscript by Ronald Kamoga, et al.
In the current manuscript, Ronald Kamoga and co-authors carried out an epidemiological investigation and analysis. They investigated the prevalence, severity of behavioral and psychological symptoms of dementia (BPSD) in patients living with dementia (PLWD), as well as the distress score in their family caregivers in South-West Uganda. Their findings have identified a very strong correlation between BPSD severity in PLWD and their caregivers’ distress. This study lays a foundation for further screening and intervention on BPSD and provides a rationale for developing possible strategies for caregiver support in Uganda.
Overall, this study is scientically sound. The data presented in the manuscript are of good quality. However, in many places of this paper, the author's description upon BPSD in patients with dementia and distress in caregivers is mixed up. There are some major issues that need to be addressed before acceptance for publication.
1) The authors need to provide basic information about the study subjects, such as multiple characteristics in patients with dementia, including activities of living, stages of dementia. These clinical features indirectly affect family caregivers’ burden as demonstrated by Bokyoung Kim, et al. (Behavioural and psychological symptoms of dementia in patients with Alzheimer’s disease and family caregiver burden: a path analysis. BMC Geriatrics. 2021, 21:160). So it would be better if the demographic and clinical characteristics of patients with dementia are included in Table 1.
2) If possible, the inclusion and exclusion criteria of patients with dementia should be provided in paragraph 2.3, titled as “Study Participants”.
3) In my understanding, only socio-demographic characteristics of caregivers are shown in Table 1. Neither cumulative number of BPSD nor associations data are presented in this Table. Therefore, I suggest the author to modify the title for Table 1 to reflect the correct contents.
4) For the title of Table 2, is frequency of Behavioral and psychological symptoms of dementia exhibited among PLWD, instead of among caregivers? Please clarify this in the Title.
5) The interpretation on “the distress score” in caregivers is quite confusing. According to the description in Lines 253-262, it seems that the caregivers’ distress exhibits the same Behavior and Psychological Symptoms as BPSD in patients with dementia. The same confusion is also shown in Lines 265-274. Please make sure the distinction between the distress in caregivers and BPSD in PLWD is stated clearly and unequivocally.
6) In Lines 265-266, it is inappropriate to draw the conclusion “A unit increase in the severity of the symptom will cause 0.82 times the worsening of the distress reported by caregivers.” Rather, the Pearson correlation coefficient 0.82 only reflects the strength of relationship between two variables, but not the degree of changing of one variable following another. The authors need to make a correction on the referred conclusion statement.
7) All tables should be cited in the manuscript.
The minor weaknesses are:
1) Too many grammatical and spelling mistakes in this paper need extensive proof reading.
2) Past tense should be used to describe the experimental results.
3) Sentences should be well organized and more concise for readers to understand.
4) The design of Table format should be rationally distributed and more aesthetical.
Round 2
Reviewer 1 Report
Most of suggestions appear to have been taken into consideration. However, there is still some part of the text noted above by this reviewer that could be explained in more detail.
